# Vegetative Propagation of *Phytophthora cinnamomi*-Tolerant Holm Oak Genotypes by Axillary Budding and Somatic Embryogenesis

**Maria Teresa Martínez** [1,†], **Francisco Javier Vieitez** [1], **Alejandro Solla** [2,‡] , **Raúl Tapias** [3,‡] , **Noelia Ramírez-Martín** [4] **and Elena Corredoira** [1,*,†]

1   Instituto de Investigaciones Agrobiológicas de Galicia (IIAG), Avd Vigo s/n, 15705 Santiago de Compostela, Spain; temar@iiag.csic.es (M.T.M.); jvieitez@iiag.csic.es (F.J.V.)
2   Faculty of Forestry, Institute for Dehesa Research (INDEHESA), University of Extremadura, Avenida Virgen del Puerto 2, 10600 Plasencia, Spain; asolla@unex.es
3   Departamento de Ciencias Agroforestales, Universidad de Huelva, ETSI Campus de La Rábida, 21819 Palos de la Frontera, Huelva, Spain; rtapias@uhu.es
4   Instituto Madrileño de Investigación y Desarrollo Rural, Agrario y Alimentario (IMIDRA), Apartado 127, 28800 Alcalá de Henares, Spain; noelia.ramirez.martin@madrid.org
*   Correspondence: elenac@iiag.csic.es
†   MT Martínez and E. Corredoira contributed equally to this work.
‡   A Solla and R Tapias contributed equally to this work.

**Abstract:** Holm oak (*Quercus ilex*) is one of the most widely distributed tree species in the Mediterranean basin. High mortality rates have been observed in holm oak populations in the southwest of the Iberian Peninsula as a result of oak decline syndrome. Selection and propagation of genotypes tolerant to this syndrome could aid the restoration of affected areas. In this article, we report micropropagation and conservation procedures based on axillary budding and somatic embryogenesis (SE) of holm oak plants, selected for their tolerance to *Phytophthora cinnamomi*—the main biotic factor responsible for oak decline. Forced shoots were obtained from potted plants of eight different genotypes, and used as stock material to establish in vitro shoot proliferation cultures. Reliable shoot proliferation was obtained in seven out the eight genotypes established in vitro, whereas multiplication rates were genotype-dependent. The highest rooting rates were obtained by culturing shoots for 24 h or 48 h on rooting induction medium containing 25 mg L$^{-1}$ indole-3-butyric acid, followed by transfer to medium supplemented with 20 μM silver thiosulphate. Axillary shoot cultures can be successful conserved by cold storage for 12 months at 4 °C under dim lighting. Shoot tips, excised from axillary shoot cultures established from tolerant plants, were used as initial explants to induce SE. Somatic embryos and/or nodular embryogenic structures were obtained on induction medium with or without indole-acetic acid 4 mg L$^{-1}$, in two out the three genotypes evaluated, and induction rates ranged between 2 and 4%. Plantlet recovery was 45% after two months cold stratification of somatic embryos and eight weeks of culture on germination medium. Vegetative propagation of *P. cinnamomi*-tolerant *Q. ilex* trees is a valuable milestone towards the restoration of disease-affected areas.

**Keywords:** axillary shoot proliferation; clonal propagation; micropropagation; *Phytophthora cinnamomi*; *Quercus ilex*; silver thiosulphate; somatic embryogenesis; disease-tolerant plants

## 1. Introduction

In the southwest Iberian Peninsula, Mediterranean oak woodlands occur as a particular anthropogenic habitat called "dehesas" in Spain or "montados" in Portugal [1]. *Dehesas* are spaced oak woodlands in which a large part of the area is occupied by grassland managed as a savannah-like ecosystem [2]. The evergreen oaks, such as holm oak (*Q. ilex* L) and cork oak (*Q. suber* L), are the main tree components of this agrosilvopastoral ecosystem and are of great socioeconomic and ecological importance in the SW Iberian Peninsula. Two subspecies of *Q. ilex*, which differ in morphology, distribution and drought stress, have been identified: *Quercus ilex* subsp. *ballota*, synonymous with subsp. *rotundifolia*, native to the southwest of the species geographic range, including the dehesas, and *Q. ilex* subsp. *ilex*, native to the north and east of the range [3]. Animals are an essential part of the dehesas, in which hunting species coexist with livestock species, such as cattle, sheep, goats, and pigs, that transform the production of grass, acorns and branches into products of premium quality and high added value (e.g., meat, including the internationally renowned acorn-fed Iberian ham, and cheeses). Another by-product of holm oak dehesas is the highly valued truffle or edible fungus [4]. The ecological importance of this ecosystem is indisputable, as it provides a multitude of advantages in terms of climate regulation, water provision, erosion control and carbon sequestration, among others [5].

Oak decline syndrome is a phenomenon of great concern and affects oak stands worldwide [6]. The sustainability of the dehesa ecosystem is threatened by oak decline syndrome (called "la seca" in Spanish). This severe disease has caused high mortality rates in holm oaks for several years and remains active at present. In most cases, disease episodes occur as the result of the interactions between different abiotic agents (severe drought, extreme temperatures, waterlogging, etc.) with and without the involvement of biotic agents. According to several authors, *Phytophthora cinnamomi* is the main biotic agent causing the disease [7,8], although *P. quercina* can also be involved [9]. Another factor threatening the sustainability of the dehesa is the low natural regeneration capacity of the woodland, mainly due to aging of the trees. The highly variable production and germination of acorns and their high predation by livestock, along with their limited dispersion, also negatively affect the natural regeneration capacity of holm oak [8,10]. Moreover, land abandonment, land-use intensification, habitat fragmentation and transformation of the land into cultivated and urban areas may also affect the viability of the dehesas [11,12]. Numerous studies have highlighted the Mediterranean basin as potentially highly vulnerable to global change. A marked decrease in precipitation at the same time as the occurrence of very high-temperature episodes can cause decline or forest dieback across an area [12,13].

Attempts have been made to control the disease in the dehesas to reduce the risk of dispersion/infectivity of *P. cinnamomi*, e.g., by applying limestone amendments to the soil [14], injecting phosphonates in the trunk [15] and biofumigating the trees [16,17]. However, limited success was achieved with all of these techniques. The use of genetically improved varieties to restore holm oak populations in affected areas is a possible alternative to chemical control of the disease [18]. Nevertheless, long-term conventional breeding programs aimed at producing *P. cinnamomi*-tolerant genotypes by controlled crossings have not been conducted [19]. In the last few years, holm oak plants tolerant to *P. cinnamomi* have been identified by the selection of surviving adult trees in oak decline hotspots. Acorns were collected from selected trees and germinated. Once the plants were established, they were repeatedly inoculated with the oomycete to confirm tolerance. Seedlings of different provenances of holm oak and cork oak collected in Huelva (Spain) were inoculated with *P. cinnamomi* to test their tolerance to the oomycete [20]. In addition, 100 pathogen-tolerant seedlings have been selected from 16 populations of *Q. ilex* located in seven Mediterranean countries [21].

Once the selected holm oak material is available, the next step involves the clonal propagation of this material to enable the capture of all gene combinations and provide strong genetic gains. Conventional clonal propagation of oaks, including holm oak, either by stooling [22] or by root cuttings [23] is very difficult and is clearly affected by the age of the ortet. Attempts to propagate

the *P. cinnamomi*-tolerant plants used in the present study by rooting cuttings were unsuccessful, regardless of the age of the mother plant (Dr A. Solla, pers. Comm.). The clonal propagation of mature elite individuals, threatened or endangered tree species, and genotypes with known resistance to pathogens is an important goal in forestry [24]. Micropropagation is a potentially valuable alternative to conventional clonal propagation, although oaks (including holm oak) have always been considered recalcitrant to in vitro tissue culture [25,26]. Axillary shoot proliferation and somatic embryogenesis (SE) are the main procedures used for micropropagation and conservation of hardwood species [27,28]. Although propagation by axillary budding has been reported for several oak species [25], to date few protocols have been developed for holm oak. In a very preliminary study, [29] described a method of culturing shoots obtained from germinated acorns, although the procedure was not clearly explained. Recently, [30] reported that the establishment and proliferation of shoot cultures from mature holm oaks, although rooting and acclimatization rates were very low.

Somatic embryogenesis is a powerful tool in clonal forestry, in the cryopreservation of valuable germplasm, and it also forms the basis of genetic transformation procedures [28]. Somatic embryos of holm oak have been initiated from zygotic embryos [31], floral tissues such as developing ovules [26] and male catkins [32], although low embryo induction rates have been obtained. There are several difficulties associated with the use of floral explants to induce SE, including the correct excision of the floral tissues to be cultured and the seasonal limitations to the availability of the material. In order to overcome these problems, [33] highlighted the induction of somatic embryos from leaves and shoot apex explants excised from in vitro shoot cultures established from holm oak trees. The advantages of using shoot proliferation cultures as a source of explants for inducing SEs include the fact that sterilization of the explants is avoided, material at the same physiological stage can be produced throughout the year, and the growing conditions of the stock material can be controlled [25].

In order to alleviate the ecological and economic impact of oak decline on holm oak populations in the Iberian Peninsula, the main goal of this study was to develop a method for the clonal multiplication of tolerant holm oak plants. Two micropropagation procedures were evaluated for this purpose: (1) Plant regeneration based on axillary budding, including optimization of the rooting step; and (2) induction of SE from explants excised from the same shoot culture lines generated in the axillary budding procedure. Conservation of shoot cultures of the valuable germplasm by cold storage was also considered.

## 2. Materials and Methods

### 2.1. Plant Material

Potted *P. cinnamomi*-tolerant holm oak plants were used as a source of explants for in vitro shoot culture establishment. Stock plants (4–5 years old) of *Quercus ilex* subsp. *Ilex* (genotypes PR6, PR8 and PR-11) and of *Quercus ilex* subsp. *Ballota* (genotype PR9) were selected by researchers at the University of Extremadura (Uex) (SW Spain) [21]. Stock plants (7 years old) of *Quercus ilex* subsp. *Ballota* (genotypes E1-1, E2-1, E3-2 and ES5-1) were selected by researchers at the University of Huelva (Uhu) (SW Spain) [20]. All tolerant plants derived from acorns were harvested in geographical areas affected by oak decline. Plants selected by the Uhu were collected in Huelva (SW Spain), whereas the Uex researchers selected the plants in different geographical areas of Spain, Portugal and Italy (see Supplementary information S1).

### 2.2. In Vitro Shoot Establishment

Genotypes PR8 and PR9 were forced to flush in a greenhouse, while genotypes PR6, PR11, E1-1, E2-1, E3-2 and ES5-1 were forced to flush in a growth cabinet at 25 °C, 80–90% relative humidity and a 16-h photoperiod (90–100 µmol m$^{-2}$ s$^{-1}$ provided by cool-white fluorescent lamps). Approximately 2–4 weeks later (Figure 1A), new actively growing shoots were collected and used as a source of explants (see Supplementary information S2). The shoots were stripped of their leaves and washed

with sterile water for 10 min before sterilization for 2 min 30 s with an aqueous solution of 0.2% sodium hypochlorite (chlorine-free) (Millipore, Merck, Burlington, MA, USA), to which 2-3 drops of Tween 80$^{®}$ were added. The shoots were then washed three times (10 min each time) with sterile water. Initial explants, consisting of 0.5–1.0 cm nodal sections, were cultured upright in culture tubes (30 × 150 mm) containing 20 mL of establishment medium consisting of Woody Plant Medium (WPM) [34] (Duchefa Biochemie, Haarlem, Netherlands) supplemented with 80 mg L$^{-1}$ ascorbic acid (Sigma-Aldrich, St. Louis, MO, USA), 30 g L$^{-1}$ sucrose (Duchefa Biochemie, Netherlands), 7 g L$^{-1}$ Vitroagar (Pronadisa, Madrid, Spain) and 0.2 mg L$^{-1}$ benzyladenine (BA) (Sigma-Aldrich, St. Louis, MO, USA) (see Supplementary information S3). The medium was adjusted to pH 5.7 and autoclaved at 115 °C for 20 min. The ascorbic acid was filter-sterilized before being added to the autoclaved medium. All cultures were maintained in a growth chamber with a 16-h photoperiod (50–60 μmol m$^{-2}$ s$^{-1}$ provided by cool-white fluorescent lamps at 25 °C (light) and 20 °C (dark) (standard conditions). After 24 h of culture, each explant was moved to the opposite side of the culture tube to prevent contact with any excreted phenolic compounds. The cultures were transferred every two weeks to a fresh medium of the same composition. After eight weeks, the percentages of explants showing contamination, necrosis and new sprouting buds (responsive rate) were recorded for each genotype.

## 2.3. In Vitro Shoot Proliferation

After the eight-week culture period, shoots longer than five mm were excised from the original explants and cultured in 0.5 L glass jars containing 70 mL of shoot proliferation medium consisting of WPM supplemented with 30 g L$^{-1}$ sucrose, 8 g L$^{-1}$ Sigma agar (A-1296; Sigma-Aldrich, St. Louis, MO, USA), 20 μM filter-sterilized silver thiosulphate (STS) (Millipore, Merck, Darmstadt, Germany) and specific BA concentrations (see Supplementary information S3). The medium was adjusted to pH 5.7 and autoclaved at 115 °C for 20 min. Shoots were transferred to fresh medium every two weeks over a six-week standard proliferation cycle, as follows—0.1 mg L$^{-1}$ BA for the first two weeks, 0.05 mg L$^{-1}$ BA for the next two weeks, and 0.01 mg L$^{-1}$ BA for the last two weeks. Subculture on this proliferation medium and with this cytokinin regime was repeated successively until the shoot cultures stabilized. The stabilization period was defined as the time required for uniform, continuous growth [35].

Once the culture was stabilized, the effect of genotype on shoot proliferation ability was evaluated. Shoots of genotypes PR8, PR9, PR11, E2-1 and E3-2 were cultured on proliferation medium for a standard, six-week proliferation cycle. At the end of this period, the following parameters were quantified: The percentage of explants that formed shoots (responsive explants), the mean number of shoots per responsive explant (shoots 0.5–1.0 cm long, and shoots >1.0 cm long) and the mean length of the longest shoot among the responsive explants. Five replicate jars, each containing seven explants (35 explants per treatment), were used and each experiment was repeated at least twice.

## 2.4. In Vitro Rooting

In the first trial, the rooting ability of the different genotypes was compared by using a rooting medium previously defined for holm oak [30]. Shoots (1.0–1.5 cm long) isolated from six-week-old shoot proliferating cultures of genotypes PR8, PR9, PR11, E2-1 and E3-2 were cultured on Murashige and Skoog medium (MS) [36] (Duchefa Biochemie, Haarlem, Netherlands) with half-strength macronutrients ($\frac{1}{2}$ MS) supplemented with 30 g L$^{-1}$ sucrose, 6 g L$^{-1}$ Vitroagar, 3 mg L$^{-1}$ indol-3-butyric acid (IBA) (Duchefa Biochemie, Haarlem, Netherlands) and 0.1 mg L$^{-1}$ α-naphthalene acetic acid (NAA) (Duchefa Biochemie, Haarlem, Netherlands). After 15 days on the rooting medium, the shoots were transferred to the same medium without auxins and supplemented with 20 μM STS.

A second experiment was carried out with shoots of genotypes PR11 and E2-1. The shoots were cultured in Gresshoff and Doy medium (GD) [37] (Duchefa Biochemie, Haarlem, Netherlands) supplemented with 30 g L$^{-1}$ sucrose, 6 g L$^{-1}$ Vitroagar and 25 mg L$^{-1}$ IBA and with macronutrients reduced to one-third strength (1/3 GD) (rooting medium). After 24 or 48 h on this rooting medium, the shoots were transferred to the same medium without IBA and supplemented with either 20 μM

STS or 0.4% activated charcoal (AC) (Sigma-Aldrich, St. Louis, MO, USA) (rooting expression medium; (see Supplementary information S3)).

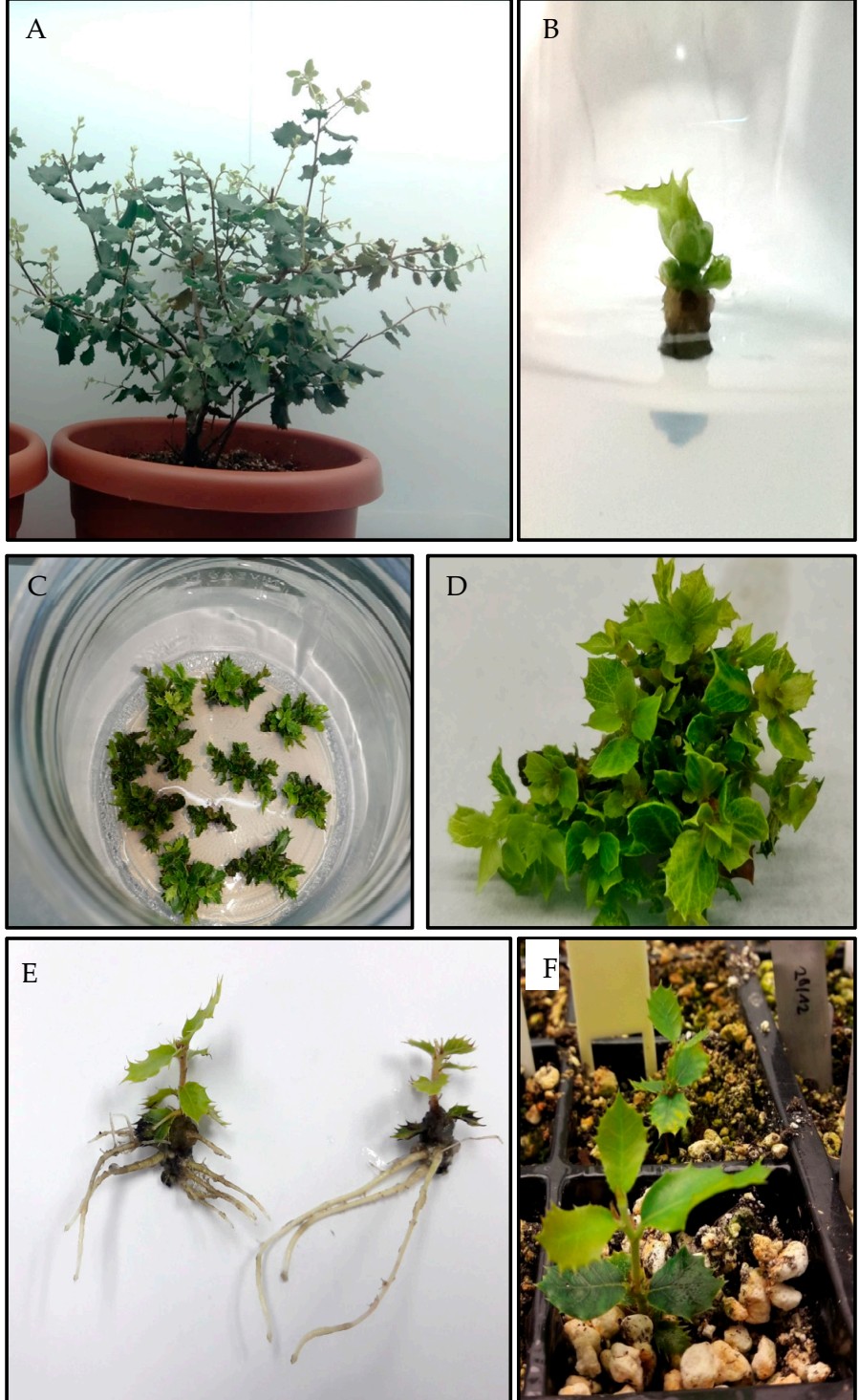

**Figure 1.** Micropropagation of tolerant holm oak plants by axillary shoot proliferation. (**A**) Forced flushing of the potted plant placed in a growth chamber. (**B**) Axillary shoot arising from a nodal explant after eight weeks of culture. (**C,D**) Shoot cultures appearance of genotype E2-1 after growth in proliferation medium. (**E**) Root development on PR11 shoots treated with 25 mg L$^{-1}$ IBA for 48 h and subsequent transfer to auxin free medium containing 20 µM STS (sterilized silver thiosulphate). (**F**) Acclimatized plants of genotype PR11 after six weeks in the growth medium.

In a third experiment, shoots of genotypes PR8, PR9 and E3-2 were cultured for 24 or 48 h in the same rooting medium defined in the second experiment, before being transferred to medium without auxin and supplemented with 20 µM STS.

In each rooting experiment, twenty-five shoots were used per genotype and treatment, and each experiment was repeated at least twice. The percentage of shoots that rooted, the mean number of roots per rooted shoot, and the length of the longest root (mm) of each shoot were determined after six weeks.

### 2.5. Cold Storage of Shoot Cultures

Basal segments with 2-3 axillary buds excised from six-week-old in vitro-shoot cultures of genotypes PR9, PR11, E2-1 and E3-2 were used as explants for in vitro conservation by cold storage. Prior to cold storage, explants were kept under standard conditions for 15 days on proliferation medium supplemented with 0.1 mg $L^{-1}$ BA. Subsequently, glass jars were placed into 340 L Sanyo Medicool Cabinets at 4 °C under dim lighting (8–10 µmol $m^{-2}$ $s^{-1}$) provided by exterior cool-white fluorescent lamps. After cold storage for 9 and 12 months, the cultures were removed from the cabinets and immediately transferred to fresh medium and cultured for four weeks in a growth chamber under standard conditions. Controls (0 months in the cold) were maintained under standard proliferation cycle conditions during the experiment (see subheading 2.3). After four weeks in the proliferation medium, the following parameters were evaluated: Recovery, defined as the percentage of cultures showing shoot proliferation after cold storage; the total number of new shoots per explant; the number of shoots of lengths 0.5–1.0 cm and >1.0 cm per explant; and the mean length of the longest shoot (mm) in the responsive explants. Five replicate jars, each containing seven explants (35 explants per treatment) were used for each cold storage period and genotype, and the experiment was repeated once.

### 2.6. Somatic Embryogenesis Procedures

Axillary shoot cultures previously established from tolerant plants of genotypes PR9, PR11 and E2-1 were used as a source of initial explants to induce somatic embryogenesis. Ten shoot tips, of length 2.0–2.5 mm and comprising the apical meristem and 2-3 leaf primordia, were excised from actively growing shoots and placed horizontally in 90 mm Petri dishes containing 25 mL of embryo induction medium consisting of MS mineral salts and vitamins, 500 mg $L^{-1}$ casein hydrolysate (Duchefa Biochemie, Haarlem, Netherlands), 30 g $L^{-1}$ sucrose and 6 g $L^{-1}$ Vitroagar supplemented with 4 mg $L^{-1}$ NAA or 4 mg $L^{-1}$ indole-acetic acid (IAA) (Duchefa Biochemie, Haarlem, Netherlands) in combination with 0.5 mg $L^{-1}$ BA. Induction medium devoid of PGRs was also tested (control treatment). Explants were cultured for two or eight weeks in darkness at 25 °C. The shoot tips were cultured for 16 weeks to a second medium consisting of induction medium without PGRs [33]. Fifty shoot tip explants were used per genotype and treatment, and the experiments were repeated at least twice. The embryogenic response was evaluated by recording the number of explants forming somatic embryos or nodular embryogenic structures (NSs).

Somatic embryo proliferation and maintenance of the embryogenic competence was achieved by repetitive embryogenesis, by culturing somatic embryos or NSs in 90 mm Petri dishes containing 25 mL of Schenk and Hildebrandt (SH) [38] medium (Duchefa Biochemie, Haarlem, Netherlands) without PGRs to yield new cycles of somatic embryos and NSs every 5–6 weeks. To determine the embryo germination and plantlet conversion ability, isolated cotyledonary somatic embryos (≥5 mm) were placed in empty Petri dishes and stored at 4 °C for two months. After cold storage, embryos were cultured in 0.5 L glass jars containing 70 mL of germination medium consisting of GD medium supplemented with 30 g $L^{-1}$ sucrose, 6 g $L^{-1}$ Vitroagar, 0.1 mg $L^{-1}$ BA and 20 µM STS. After eight weeks, the germination response was determined by recording the number of embryos with roots only and the number of embryos that converted into plants (both shoot and root development ≥5 mm). Five replicate jars, each containing six explants (30 explants per treatment) were used to evaluate plant conversion ability.

### 2.7. Statistical Analysis

The statistical analysis was conducted using SPSS for Windows (version 23, Chicago, IL, USA). Data were statistically analyzed by one-way factorial analysis of variance (ANOVA I) for the experiments shown in Tables 2, 3 and 6, and by two-way factorial analysis of variance (ANOVA II) for the experiments shown in Tables 4 and 5. In ANOVA analysis, percentage data were subjected to arcsine transformation prior to analysis, but the data presented in the tables are untransformed. In Table 7, data were analyzed by the Chi squared ($\chi^2$) test from a contingency table.

## 3. Results

### 3.1. In Vitro Shoot Establishment

The stock tolerant plants of all eight genotypes evaluated started to flush after 2–4 weeks, although there were differences in the number of new flushed shoots produced. The Uex plant selection produced more shoots than the Uhu selection. Mean shoot length of the flushed shoots did not vary between the different stock plants, ranging from 2.5 to 4.5 cm in most of the genotypes, except for the E1-1 tolerant plant, which produced the longest shoots (Table 1). High level contamination occurred in initial cultures derived from plants grown in the greenhouse (genotypes PR8 and PR9 with incidences of 71.4 and 65.8% contamination, respectively), relative to plants forced to flush in the growth cabinet (0 to 4.2% contamination). After inoculation in the culture medium, the explants secreted within 24 h brown exudates, probably consisting of phenolic compounds. Ascorbic acid was added to the culture medium, and the explants were moved to the opposite side of the test tube to mitigate this negative effect. However, the frequency of necrotic explants was generally high and seemingly genotype-dependent, with values above 50% in four out of the eight genotypes studied.

**Table 1.** Flushing capacity of different potted tolerant *Quercus ilex* plants and in vitro establishment of nodal explants excised from flushed shoots.

| Genotype | | Responsiveness to Flushing | | Responsiveness to In Vitro Establishment | | | |
|---|---|---|---|---|---|---|---|
| | | Shoots Harvested per Plant (N°) | Length of Shoots (cm) | Initial Nodal Explants (N°) | Contamination Rates (%) [1] | Necrosis Rates (%) [1] | Responsive Explants (%) [1,2] |
| UEx Selection | PR6 [3,5] | 30 | 2.98 ± 1.1 | 62 | 1.6 | 30.6 | 4.8 |
| | PR8 [4,5] | - | - | 35 | 71.4 | 70.0 | 10.0 |
| | PR9 [4,6] | - | - | 38 | 65.8 | 23.1 | 23.1 |
| | PR11 [3,5] | 29 | 3.45 ± 1.4 | 46 | 2.2 | 54.3 | 54.3 |
| UHu Selection | E1-1 [3,6] | 9 | 8.36 ± 3.19 | 48 | 4.2 | 81.3 | 14.6 |
| | E2-1 [3,6] | 16 | 2.53 ± 1.36 | 40 | 2.5 | 62.5 | 35.0 |
| | E3-2 [3,6] | 13 | 2.92 ± 1.92 | 25 | 4.0 | 24.0 | 72.0 |
| | ES5-1 [3,6] | 14 | 4.52 ± 1.97 | 41 | 0.00 | 26.8 | 73.2 |

[1] Evaluated after eight weeks of culture. [2] Explants with sprouting buds after eight weeks of culture. [3] Flushing induced in a climatic cabinet. [4] Flushing induced in a greenhouse. [5] *Q. ilex* subsp ilex. [6] *Q. ilex* subsp ballota. Uex: University of Extremadura; Uhu: University of Huelva.

The flushing capacity of stock plants, in terms of shoot number and shoot length, was not related to the in vitro establishment ability in terms of the bud sprouting response of initial explants (Table 1). Although buds sprouted in all the genotypes tested (Figure 1B), there were marked differences in the response frequency, and the highest value was obtained with ES5-1 material (73.2% sprouting). Establishment of shoot cultures appeared to be influenced by the origin of plants, as the total bud sprouting response was higher in the Uhu selection than in the Uex selection.

At the end of the eight-week-long culture establishment period, shoots longer than 0.5 cm, which had developed from original explants, were isolated and successively cultured on proliferation

medium yielding stable shoot cultures. The time required to obtain uniform shoot growth ranged from 3–14 months in the different genotypes. Stable cultures were obtained in all the four genotypes from the Uex selection and in three out of the four genotypes from the Uhu selection, as ES5-1 genotype material was lost as shoots became hyperhydric during successive subculture.

### 3.2. In Vitro Shoot Proliferation

After stabilization, shoots were successfully subcultured several times on the proliferation medium to produce a sufficient number of shoots for use in the shoot proliferation experiment, which was carried out with the five genotypes, shown in Table 2.

**Table 2.** Influence of genotype on shoot proliferation in cultures derived from five tolerant *Quercus ilex* genotypes.

| Genotype | Responsive Explants (%) | Shoot Number 0.5–1.0 cm | Shoot Number >1 cm | Total Shoot Number | Longest Shoot Length (mm) |
|---|---|---|---|---|---|
| PR8 | $100.0 \pm 0.0$ | $1.2 \pm 0.1$ | $3.5 \pm 0.5$ | $4.7 \pm 0.5$ | $13.1 \pm 0.8$ |
| PR9 | $100.0 \pm 0.0$ | $2.7 \pm 0.3$ | $4.3 \pm 0.3$ | $7.0 \pm 0.5$ | $17.1 \pm 0.6$ |
| PR11 | $100.0 \pm 0.0$ | $2.4 \pm 0.4$ | $7.0 \pm 0.5$ | $9.4 \pm 0.5$ | $21.2 \pm 1.6$ |
| E2-1 | $100.0 \pm 0.0$ | $6.0 \pm 0.5$ | $5.1 \pm 0.4$ | $11.1 \pm 0.7$ | $9.7 \pm 0.6$ |
| E3-2 | $100.0 \pm 0.0$ | $1.2 \pm 0.2$ | $2.8 \pm 0.2$ | $4.0 \pm 0.4$ | $17.7 \pm 1.0$ |
| | ns | 0.001 *** | 0.001 *** | 0.001 *** | 0.001 *** |

Data represent means ± standard error of ten replicate jars with seven explants per jar. ANOVA I significance levels are shown for each parameter. ns: not significant; *** significant differences at 99.9% ($p \leq 0.001$).

Shoot proliferation was obtained in all the five genotypes evaluated, in which 100% of explants showed a positive response by developing shoots at least 0.5 cm long (Figure 1C,D). However, significant genotypic differences ($p \leq 0.001$) were observed in all of the other parameters evaluated (Table 2). With the exception of the E2-1 material, the number of shoots per explant > 1 cm was greater than those of 0.5–1 cm among the genotypes. The longest shoot length per explant ranged from 9.7 to 21.2 mm, and the highest value corresponded to the PR11 material. Genotype PR11 also produced the largest number of shoots > 1 cm, whereas the E2-1 material produced the shortest shoots, although the greatest total number of shoots (Table 2).

### 3.3. In Vitro Shoot Rooting

In an initial experiment, the rooting ability of the five genotypes used in the shoot proliferation experiments was evaluated after culture in rooting medium containing 3 mg L$^{-1}$ IBA and 0.1 mg L$^{-1}$ NAA for 15 days (Table 3). Rooting frequency and root length were significantly ($p \leq 0.001$) influenced by the genotype. Genotype PR11 yielded the highest rooting frequency and the longest roots (58.3% and 110.7 mm, respectively), whereas the other four genotypes yielded significantly lower rooting rates (0–12%).

**Table 3.** Rooting ability of shoots derived from five tolerant *Quercus ilex* genotypes.

| Genotype | Rooting (%) | Root Number | Longest Root Length (mm) |
|---|---|---|---|
| PR8 | $4.0 \pm 4.0$ | $2.0 \pm 0.0$ | $10.0 \pm 0.0$ |
| PR9 | $12.0 \pm 4.4$ | $1.3 \pm 0.2$ | $19.4 \pm 1.8$ |
| PR11 | $58.3 \pm 4.2$ | $2.4 \pm 0.4$ | $110.7 \pm 8.7$ |
| E2-1 | $4.0 \pm 4.0$ | $1.0 \pm 0.0$ | $12.0 \pm 0.0$ |
| E3-2 | $0.0 \pm 0.0$ | - | - |
| | 0.001 *** | 0.257 ns | 0.001 *** |

Data represent means ± standard error of five replicate jars with five explants per jar. Rooting was induced by culture of shoots in $\frac{1}{2}$ MS medium supplemented with 3 mg L$^{-1}$ IBA plus 0.1 mg L$^{-1}$ NAA for 15 days and subsequently transferred to $\frac{1}{2}$ MS medium devoid of auxin and containing 20 μM STS. ANOVA I significance levels are shown for each parameter. Ns: not significant; *** significant differences at 99.9% ($p \leq 0.001$).

In order to improve the rooting capacity, one genotype from each origin (PR11 and E2-1) was used to investigate the application for 24 or 48 h of a rooting medium supplemented with 25 mg L$^{-1}$ IBA, with subsequent transfer to a root expression medium supplemented with STS or AC (Table 4). Rooting was achieved in both genotypes, confirming that PR-11 also showed a higher rooting capacity than E2-1 when the auxin treatments were applied (see also Table 3).

In the PR11 genotype, the rooting frequency was significantly affected ($p \leq 0.01$) by auxin exposure, and the best results were obtained with 24 h IBA treatment and transfer to a medium supplemented with STS or AC (Table 4). When AC was used in the 24 h IBA treatment, the rooting frequency reached 96%. However, when AC was used after 48 h of exposure, the rooting frequency equaled that obtained in the control treatment. Incorporation of STS in the root expression medium significantly ($p \leq 0.01$) increased the rooting frequency, with values ranging from 84–96% regardless of IBA exposure. The mean root number was also significantly affected by the two factors evaluated, but not by their interaction (Table 4). The best results were obtained with 24 h IBA and the inclusion of STS in the root expression medium. Root length was not affected by any of the factors tested (Table 4). In the E2-1 genotype, both factors and their interaction significantly influenced the rooting rate. The highest rooting frequency (36%) was obtained with 48 h IBA treatment and STS, while the addition of AC had a negative effect on rooting with the two periods of auxin exposure. Although the root number was not affected by IBA treatment, the presence of STS increased the root number significantly ($p \leq 0.01$). By contrast, the effect of both factors on root length is not clear.

For both genotypes, inclusion of the STS in root expression medium greatly increased both the rooting frequency and root number (Table 4; Figure 1E). The effect of auxin exposure time varied depending on the genotype, as the best response in PR11 was obtained with exposure to IBA for 24 h, whereas the best treatment for the E2-1 genotype was exposure to IBA for 48 h.

According to the results obtained in the above experiment, the rooting capacity of genotypes PR8, PR9 and E3-2 was also investigated by comparing the two IBA exposure treatments and successive transfer to a root expression medium containing 20 μM STS (Table 5). Rooting frequency was significantly ($p \leq 0.001$) affected by the genotype: Rooting was achieved in two out the three genotypes evaluated (genotype E3-2 did not form roots). Although the rooting frequency was highest after 48 h IBA treatment in PR8 and PR9, this parameter was not significantly affected by culture on IBA medium. The rooting frequency was also higher than previously obtained with 3 mg L$^{-1}$ IBA medium for 15 days (Tables 3 and 5). Mean root number was significantly ($p \leq 0.05$) influenced by IBA treatment, and root production was highest after 48 h IBA exposure. Finally, root length was not affected by either the genotype or the auxin treatment, and no significant interaction was observed for any of the parameters evaluated.

Rooted plantlets were transferred to pots containing a mixture of garden soil (Terrahum compost, Germany) and perlite (1:2), and were placed in the growth cabinet for acclimatization. After hardening off for 6–8 weeks, 20% of the plants survived, and growth was resumed (Figure 1F).

**Table 4.** Effect of indol-3-butiric acid (IBA) treatment period and the addition of 20 μM silver tiosulphate (STS) or 0.4% activated charcoal (AC) on the rooting ability of shoots from two tolerant *Quercus ilex* genotypes.

| IBA Treatment Period | Rooting (%) | | | Root Number | | | Longest Root Length (mm) | | |
|---|---|---|---|---|---|---|---|---|---|
| **PR11 (Uex)** | **Control** | **+STS** | **+AC** | **Control** | **+STS** | **+AC** | **Control** | **+STS** | **+AC** |
| 24 h | 76.0 ± 6.7 | 96.0 ± 3.6 | 96.0 ± 3.6 | 2.9 ± 0.3 | 4.6 ± 0.4 | 3.1 ± 0.3 | 31.7 ± 1.9 | 33.0 ± 1.1 | 27.0 ± 5.2 |
| 48 h | 76.0 ± 8.0 | 84.0 ± 3.6 | 76.0 ± 6.7 | 3.1 ± 0.5 | 3.7 ± 0.7 | 2.4 ± 0.3 | 31.4 ± 5.9 | 37.6 ± 4.8 | 28.4 ± 9.5 |
| Effect | | | | | | | | | |
| IBA period (A) | 0.003 ** | | | 0.047 * | | | 0.531 ns | | |
| Root expression medium (B) | 0.004 ** | | | 0.001 *** | | | 0.125 ns | | |
| A × B | 0.234 ns | | | 0.155 ns | | | 0.814 ns | | |
| **E2-1 (Uhu)** | **Control** | **+STS** | **+AC** | **Control** | **+STS** | **+AC** | **Control** | **+STS** | **+AC** |
| 24 h | 8.0 ± 4.4 | 8.0 ± 4.4 | 6.0 ± 2.9 | 1.5 ± 0.2 | 3.0 ± 0.5 | 1.0 ± 0.0 | 25.0 ± 2.2 | 12.5 ± 1.1 | 10.8 ± 0.4 |
| 48 h | 20.0 ± 8.0 | 36.0 ± 10.5 | 4.0 ± 2.5 | 2.5 ± 0.3 | 3.8 ± 1.1 | 1.0 ± 0.0 | 16.3 ± 2.0 | 17.0 ± 2.0 | 13.0 ± 0.3 |
| Effect | | | | | | | | | |
| IBA time (A) | 0.001 *** | | | 0.376 ns | | | 0.712 ns | | |
| Root expression medium (B) | 0.003 ** | | | 0.032 * | | | 0.002 ** | | |
| A × B | 0.008 ** | | | 0.831 ns | | | 0.005 ** | | |

Data represent means ± standard error of ten replicate jars with five explants per jar. Rooting was induced by shoots cultured in 1/3 GD medium supplemented with 25 mg L$^{-1}$ IBA. After auxin treatment, shoots were transferred to root expression medium consisting of 1/3 GD medium devoid of IBA and supplemented with 20 μM STS (+STS) or 0.4% AC (+AC). Control was root expression medium without auxin, STS or AC. ANOVA II significance levels are shown for each effect and parameter. Ns: not significant; * significant differences at 95% ($p \leq 0.05$); ** significant differences at 99% ($p \leq 0.01$); *** significant differences at 99.9% ($p \leq 0.001$). h: hours.

**Table 5.** Effect of indol-3-butiric acid (IBA) treatment period and genotype on the rooting ability of shoots from three tolerant *Quercus ilex* genotypes.

| Genotype | Rooting (%) | | Root Number | | Longest Root Length (mm) | |
|---|---|---|---|---|---|---|
| | IBA 24 h | IBA 48 h | IBA 24 h | IBA 48 h | IBA 24 h | IBA 48 h |
| PR8 | 16.0 ± 3.6 | 20.0 ± 5.8 | 1.3 ± 0.2 | 1.8 ± 0.4 | 6.3 ± 1.0 | 6.3 ± 1.0 |
| PR9 | 12.0 ± 4.4 | 16.0 ± 6.8 | 1.0 ± 0.0 | 2.3 ± 0.1 | 26.7 ± 4.5 | 23.3 ± 2.2 |
| E3-2 | 0.0 ± 0.0 | 0.0 ± 0.0 | - | - | - | - |
| Effect | | | | | | |
| Genotype (A) | 0.001 *** | | 0.320 ns | | 0.066 ns | |
| IBA period (B) | 0.704 ns | | 0.03 * | | 0.708 ns | |
| A x B | 0.964 ns | | 0.405 ns | | 0.071 ns | |

Data represent means ± standard error of five replicate jars with five explants per jar. Rooting was induced by culturing the shoots in 1/3 GD medium supplemented with 25 mg L$^{-1}$ IBA and subsequently transferred to 1/3 GD medium devoid of IBA and supplemented with 20 µM STS. ANOVA II significance levels are shown for each effect and parameter. Ns: not significant; * significant differences at 95% ($p \leq 0.05$); *** significant differences at 99.9% ($p \leq 0.001$). h: hours.

## 3.4. Conservation of Axillary Shoots by Cold Storage

Holm oak shoot cultures can be successfully stored in vitro at 4 °C for at least 12 months, with relatively high recovery rates (83–100%) (Table 6). Cold storage moderately affected shoot morphology, across all genotypes, as the shoots turned yellowish green due to the reduction in light intensity applied during storage (Figure 2A). Necrosis of shoot tips and leaves occurred only in the cultures stored for 12 months. However, after the cultures were transferred to standard conditions for four weeks, no morphological differences were observed relative to the controls (Figure 2B).

**Table 6.** Effect of cold storage period on recovery and proliferation capacity of shoot cultures derived from four tolerant *Q. ilex* genotypes.

| Genotype | Storage Period (Months) | Recovery (%) [1] | Total Shoot Number | Shoot Number 0.5–1.0 cm | Shoot Number >1 cm | Longest Shoot Length (mm) |
|---|---|---|---|---|---|---|
| PR9 | 0 | 100.0 ± 0.0 | 6.6 ± 0.6 | 2.8 ± 0.4 | 3.8 ± 0.2 | 17.2 ± 0.6 |
| | 9 | 100.0 ± 0.0 | 6,4 ± 0.3 | 1.3 ± 0.3 | 5.1 ± 0.6 | 14.7 ± 0.8 |
| | 12 | 97.1 ± 2.5 | 6.3 ± 0.2 | 2.0 ± 0.2 | 4.3 ± 0.3 | 16.1 ± 0.6 |
| ANOVA I | | 0.397 ns | 0.936 ns | 0.095 ns | 0.158 ns | 0.161 ns |
| PR11 | 0 | 100.0 ± 0.0 | 9.7 ± 0.4 | 2.6 ± 0.5 | 7.1 ± 0.3 | 18.5 ± 1.6 |
| | 9 | 100.0 ± 0.0 | 8.2 ± 0.8 | 2.9 ± 0.3 | 5.3 ± 0.6 | 15.4 ± 0.7 |
| | 12 | 82.9 ± 6.2 | 6.4 ± 0.7 | 2.4 ± 0.3 | 4.0 ± 0.8 | 18.1 ± 1.1 |
| ANOVA I | | 0.002 ** | 0.028 * | 0.753 ns | 0.025 * | 0.411 ns |
| E2-1 | 0 | 100.0 ± 0.0 | 10.5 ± 0.6 | 5.5 ± 0.3 | 5.0 ± 0.5 | 9.8 ± 0.5 |
| | 9 | 100.0 ± 0.0 | 9.2 ± 0.5 | 2.1 ± 0.3 | 7.1 ± 0.4 | 13.7 ± 0.7 |
| | 12 | 100.0 ± 0.0 | 6.7 ± 0.6 | 3.2 ± 0.2 | 3.5 ± 0.5 | 16.7 ± 1.2 |
| ANOVA I | | 0.397 ns | 0.008 ** | 0.0001 *** | 0.003 ** | 0.001 *** |
| E3-2 | 0 | 100.0 ± 0.0 | 6,9 ± 0.2 | 2.0 ± 0.3 | 4.9 ± 0.3 | 17.4 ± 0.6 |
| | 9 | 100.0 ± 0.0 | 6.7 ± 0.8 | 1.4 ± 0.2 | 5.3 ± 0.9 | 17.7 ± 1.0 |
| | 12 | 91.4 ± 7.6 | 5.7 ± 0.4 | 1.3 ± 0.2 | 4.4 ± 0.5 | 26.0 ± 2.5 |
| ANOVA I | | 0.397 ns | 0.371 ns | 0.292 ns | 0.689 ns | 0.035 * |

[1] Recovery defined as percentage of shoots showing growth after storage for 0, 9 or 12 months at 4 °C and subsequent culture under standard conditions. Data represent means ± error standard of five replicate jars with seven shoot explants per jar. ANOVA I significance levels are shown for each genotype and parameter. Ns: not significant; * significant differences at 95% ($p \leq 0.05$); ** significant differences at 99% ($p \leq 0.01$); *** significant differences at 99.9% ($p \leq 0.001$).

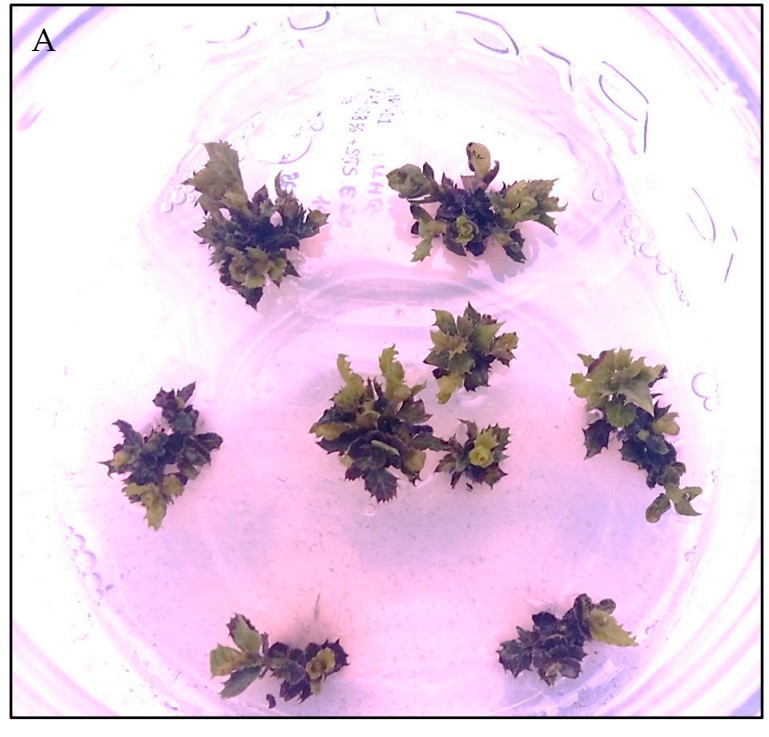

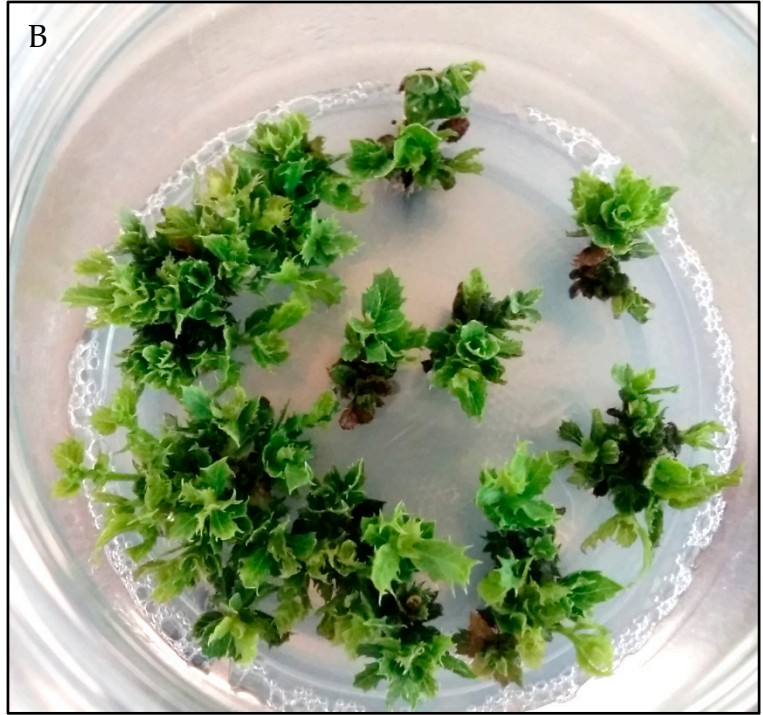

**Figure 2.** Cold storage of shoot cultures from holm oak tolerant plants. (**A**) Shoot explants after nine months of cold storage on proliferation medium under dim lighting. (**B**) Shoot recovery after nine months of cold storage, and regrowth after four weeks in culture under standard conditions.

The storage only significantly affected ($p \leq 0.01$) the recovery rate of PR11 cultures if shoots were stored for 12 months (Table 6). Total shoot number decreased as cold storage time increased in all genotypes, and the differences were significant in PR11 ($p \leq 0.05$) and E2-1 ($p \leq 0.01$). A large amount of variation in the number of shoots 0.5–1 cm in length and the number of shoots longer than 1 cm was observed for all four genotypes, while these two parameters were only significantly affected by the

storage period in E2-1 cultures. The longest shoot length was significantly affected by storage time in E2-1 and E3-2 genotypes ($p \leq 0.001$ and $p \leq 0.05$, respectively), with longest shoots recorded in cultures stored for 12 months (Table 6). However, this trend was not observed in the PR9 and PR11 genotypes.

### 3.5. Somatic Embryogenesis

In all three genotypes evaluated, the first response observed in shoot apex explants during culture for eight weeks on induction medium supplemented with NAA or IAA medium was slight callusing. However, less callus was formed when the explants were cultured on auxin induction medium for two weeks or on control medium devoid of PGRs. The slight callusing occurred in the axillary zone of most external leaf primordia and at the cut end on the shoot tip explants. Generally, NSs and/or somatic embryos began to appear after 2–3 months of culture on medium devoid of PGRs (Figure 3A,B), although the embryogenic response was also extended up to four months of culture. Somatic embryos and NSs were mainly produced from the small amount of callus tissue that formed in leaf axial zones of the explants. Nodular structures, also referred to as proembryogenic masses (PEMs), were smooth, creamy, translucent and round-oblong, whereas somatic embryos were clearly bipolar showed the typical appearance of a somatic dicotyledonous embryo.

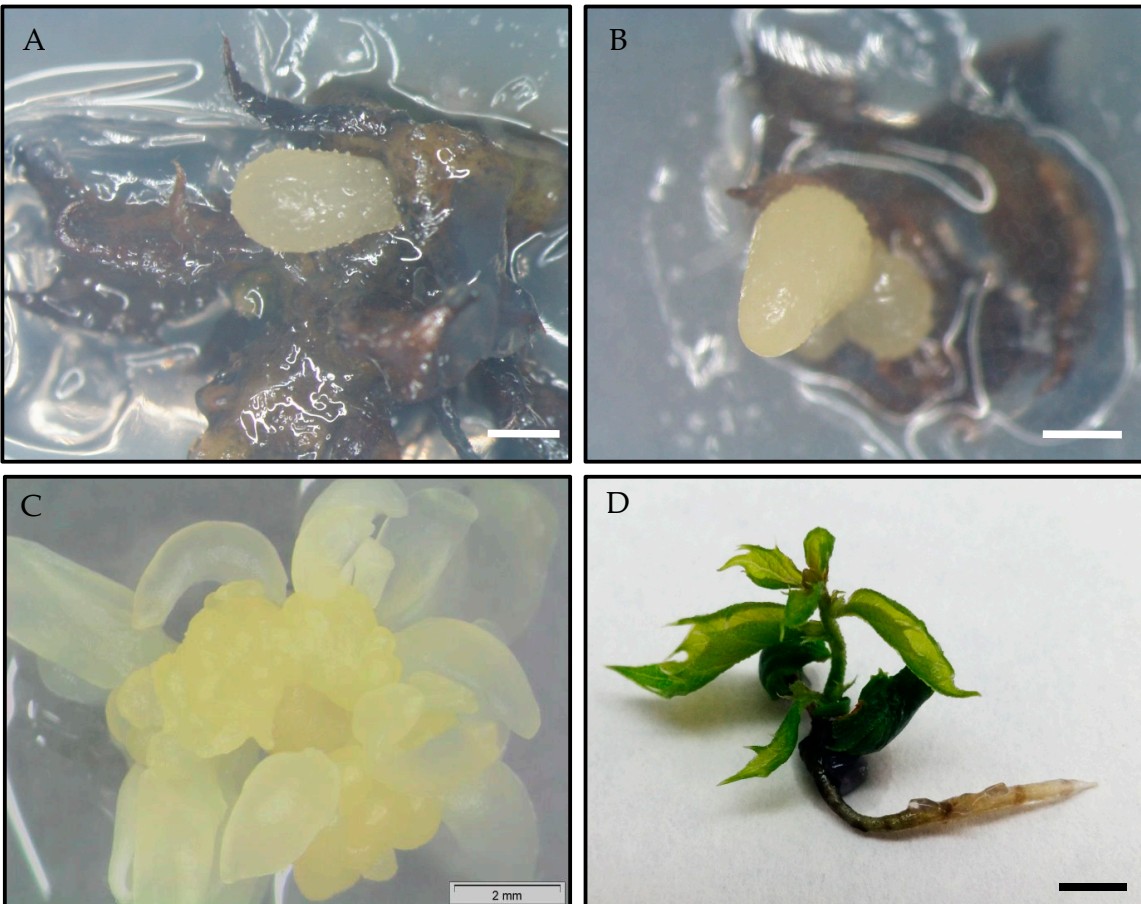

**Figure 3.** Somatic embryogenesis from shoot apex explants excised from axillary shoot cultures derived from disease-tolerant holm oak plants. (**A**,**B**). Somatic embryos initiated in shoot apex explants of genotype PR9. (**C**). Repetitive embryogenesis after six weeks of subculture of embryogenic nodular structures (NSs) on proliferation medium (**D**). Plant regeneration from stratified somatic embryos after six weeks on germination medium. Scale bar: (**A**,**B**) = 1 mm; (**D**) = 1 cm.

The SE response was affected by the genotype and the induction treatment, but the differences were not significant (Table 7). Embryogenic cultures were initiated in explants of genotypes E2-1 and

PR9, but no embryogenic response was obtained in PR11 material. Regarding the different induction media tested, NAA supplemented medium proved unsuccessful for initiating SE in any of the evaluated genotypes, in contrast to IAA medium, which was effective in E2-1 and PR9 material. Somatic embryos were generated following culture for two weeks on IAA medium in E2-1 and PR9 genotypes, and the latter were also responsive to culture on control induction medium (devoid of PGRs). No embryogenic response was obtained in explants maintained for eight weeks in IAA medium. Relatively low SE induction rates were produced, ranging between 2–4%, with the highest frequency (4%) achieved in PR9 explants. Only one somatic embryo/nodular structure was generally formed per embryogenic explant (Figure 3A,B), but the number of somatic structures occasionally increased without any clear relationship with the embryo induction treatment.

**Table 7.** Effect of genotype and auxin treatment on embryogenic response in shoot apex explants derived from axillary shoot cultures of three tolerant *Quercus ilex* genotypes.

| Genotype | Somatic Embryogenesis (%) | | | | |
| | IAA 4 (mg L$^{-1}$) | | NAA 4 (mg L$^{-1}$) | | Control |
| | 2 w | 8 w | 2 w | 8 w | All time |
|---|---|---|---|---|---|
| E2-1 | 2.0 | 0.0 | 0.0 | 0.0 | 0.0 |
| PR9 | 2.0 | 0.0 | 0.0 | 0.0 | 4.0 |
| PR11 | 0.0 | 0.0 | 0.0 | 0.0 | 0.0 |
| $\chi^2$ | ns | ns | ns | ns | ns |

Induction medium consisting of MS medium, 500 mg L$^{-1}$ casein hydrolysate, 6 g L$^{-1}$ Vitro agar, 30 g L$^{-1}$ sucrose with (4 mg L$^{-1}$ IAA or 4 mg L$^{-1}$ NAA in combination with 0.5 mg L$^{-1}$ BA) or without PGRs (control). BA, benzyaminopurine; IAA, indole-3-acetic acid; NAA, $\alpha$-naphthalene acetic acid; ns, not significant; $\chi^2$ Chi squared.

Somatic embryos and NSs could easily be separated from the original explants and multiplied by repetitive embryogenesis, to establish different embryogenic lines of the two responsive genotypes. Maintenance of embryogenic competence was initially problematical, and several lines were lost after a small number of subcultures on embryo proliferation medium. The developmental stage of the embryogenic explants to be subcultured in proliferation medium influenced embryo production and differentiation of new secondary embryos. Thus, torpedo or early cotyledonary-stage somatic embryos frequently increased in size during development, and were thus, unable to generate new secondary embryos, whereas later cotyledonay-stage usually germinated. By contrast, NSs did not directly progress towards the successive developmental stages, but gave rise to secondary embryos or other NSs. These structures were the most effective for subculturing on proliferation medium for long-term maintenance of the embryogenic capacity (Figure 3C). Approximately 80–90% of NSs responded by producing secondary embryos and other nodular structures with a mean number of 5 to 10 embryos per subcultured explant.

When cold-treated holm oak somatic embryos at cotyledonary-stage were transferred to the germination medium, greening and an increase in the size of cotyledons and hypocotyl elongation occurred within the first two weeks of culture. Whereas, epicotyl growth became evident after 2–3 weeks of culture. After eight weeks, most of the embryos had germinated, giving rise to approximately 50% embryos exhibiting only root development, in addition to 45% embryos that converted into plants with simultaneous shoot and root development (Figure 3D).

## 4. Discussion

The main challenge of the present study was to evaluate the possibility of using biotechnological approaches to produce clonal plants of selected *P. cinnamomi*-tolerant genotypes of holm oak. Biotechnological techniques are potentially useful for propagating tolerant plants of this species, considering the lack of rooting response of cuttings isolated from the stock plants in conventional

propagation procedures. Propagation of *P. cinnamomi*-tolerant *Q. ilex* trees is required to provide *dehesa* owners with appropriate plant material for restoring this valuable and currently threatened ecosystem.

This is the first study reporting axillary shoot proliferation and conservation by slow growth storage in holm oak plants that are tolerant to the root-rot pathogen *P. cinnamomi*. Previous studies have addressed micropropagation by axillary budding of *Quercus* species [25]. However, very few studies have been carried out with *Q. ilex,* and the results so far are rather inconclusive, illustrating the difficulties encountered regarding the micropropagation of holm oak [29,30].

As expected, contamination of initial cultures was higher (65.8–71.4%) in explants derived from plants forced to flush in a greenhouse than in those flushed in a growth cabinet (0.0–4.2%). In addition, induced sprouting in a climatic chamber enabled the use of a lower concentration of hypochlorite for a short time, leading to a decrease in the proportion of explants showing signs of contamination and browning. Initiation of in vitro cultures of holm oak under controlled conditions is, therefore, highly recommended. The percentage of responsive explants with sprouting buds after culture for eight weeks clearly depends on the genotype, but there were no differences between the two subspecies tested (*ballota* and *ilex*). The effect of genotype on different morphogenetic responses, including shoot culture initiation and proliferation, has been reported in a large number of woody species, including white oak [39]. In the aforementioned study, axillary shoot cultures were established with six out of eight genotypes, of age 6–7 years, with marked genotypic differences in shoot initiation and proliferation rates [39]. Although 0.5 mg $L^{-1}$ BA was necessary for initial stimulation and sprouting of nodal segments derived from mature holm oak trees [30], this concentration was detrimental (producing hyperhydric shoots) to the tolerant holm oak material. A lower BA concentration (0.2 mg $L^{-1}$) proved optimal for the development of vigorous shoots in the juvenile material used in the present study. Shoot proliferation cultures were stabilized in seven of the eight genotypes tested. Stabilization took longer than in other oak species [39]. Unfortunately, genotype ES5-1, which produced a high percentage of responsive explants (73.2%), was lost in subsequent subcultures, as shoots acquired a succulent, hyperhydric appearance.

Axillary shoot proliferation was successfully promoted by reducing the BA concentration and by transfer of cultures to a fresh medium every two weeks over a six-week subculture cycle. A similar subculture regime has also been used for the proliferation of other Fagaceae species [39,40], including holm oak [30]. In these studies, zeatin was used in combination with BA, but was not necessary for proliferation of the juvenile holm oak material in the present study (data not shown) or for *Cedrus libani* [41]. Total shoot number obtained in holm oak was higher in comparison to shoot proliferation cultures of different American oak species [39], although the shoots were shorter. This finding may be related to the typical slow growth habit of holm oak, characterized by episodic growth and development of a large number of axillary shoots with short internodes.

Holm oak is considered a difficult-to-root species, which is one of the main constraints limiting vegetative propagation. Juvenile plants of woody species generally display a higher rooting ability than mature material. However, this is not the case in holm oak, in which rooting frequencies below 20% were recorded in conventional cutting propagation from very juvenile seven-month-old stock plants [23]. In the present study, rooting was achieved in four out the five genotypes tested. Although tolerant stock plants were 4–7 years old, the rooting frequency was only acceptable in one genotype (96%), as it was below 50% in the other genotypes. However, better results were achieved than those reported by [30] with mature material, probably due to the auxin treatment, inclusion of STS in the root expression medium and also the genotype effect. As observed in the culture establishment and shoot proliferation steps, the rooting process is clearly genotype-dependent, although there was no evidence of a relationship between the two subspecies studied. Genotypes PR8 and PR11, belonging to the subspecies *ilex*, showed marked differences in rooting ability (20% and 96%, respectively). We also found that rooting frequency was higher when the microshoots were transferred (after auxin treatment) to a root expression medium containing 20 μM STS (Table 4). Silver ions in the form of STS can modulate the deleterious effects of ethylene—a gaseous plant regulator that can be synthesized during

in vitro tissue culture [42], and that negatively affects different morphogenetic processes, including shoot proliferation and genetic transformation ability [43–45]. The role of ethylene in the rooting step is unclear, as it seems to be species-dependent [46], and can promote [47,48] or reduce [49] rooting capacity. Regardless of the role of this plant regulator, the application of STS to holm oak microcuttings seems to improve rooting rates, particularly in the PR11 and E2-1 genotypes, although a larger number of genotypes should be tested to clarify this aspect. By contrast, the addition of AC to the root expression medium in holm oak had a detrimental effect on the rooting ability. However, the addition of AC promoted the rooting frequency in avocado shoots [50] and in pedunculate oak [51]. Activate charcoal adsorbs inhibitory compounds present in the culture medium, such as toxic metabolites, phenolic exudates, metal ions, and PGRs [52]. Unfortunately, only 20% of rooted shoots survived as viable plants after hardening off, thus limiting the commercial application of the procedure. Handling and technical requirements should be improved in order to increase the efficiency of the acclimatization step.

The findings of the rooting experiments showed the difficulty in optimizing the procedure. Specific modifications to the general protocol described are necessary to produce whole plants in non-responsive and recalcitrant genotypes of the species.

Somatic embryogenesis has recently been studied in holm oak to define a reliable system for mass clonal propagation, as well as for the use of somatic embryos as target material in cryopreservation and genetic transformation experiments [19,53]. In the present study, SE was induced in two out the three genotypes evaluated. Interestingly, the two responsive genotypes, PR9 and E2-1, belong to the subspecies *ballota*. Genotype, explant type and PGRs added to the culture medium are considered crucial factors in SE induction [28].

Studies on SE induction in conifer species [54] seem to indicate that the induction rate is a highly heritable trait, thus leading to the possibility of producing families with a reasonably high SE initiation capacity through conventional breeding [55]. This type of study is scarce in hardwood species. In one of the few reported studies on the subject, [56] tested SE induction in 13 open-pollinated families of eucalypts, concluding that the induction step is under the control of additive genetic effects, and that a large number of maternal parents should be screened for use in controlled pollination. This approach could be tested in unresponsive hardwood genotypes. On the other hand, indirect evidence of the effect of the genotype on SE process in hardwoods is well documented [28]. However, genotypes can respond differently under different culture conditions. For example, NAA has been considered the standard PGR for inducing SE in immature zygotic embryos of *Eucalyptus globulus* [57]. Application of the same protocol to shoot apex and expanded leaf explants from two genotypes of mature *E. globulus* trees did not induce SE, whereas a positive response was obtained after modification of the culture medium, by replacing NAA with picloram [58]. The lack of response of holm oak genotype PR11 observed in the present study may be due to inappropriate culture conditions, regardless of the genetic profile. This may also apply to the unresponsive genotypes in in vitro axillary shoot establishment and rooting mentioned in the present study.

Somatic embryos induced from non-zygotic embryos, leaf, shoot apex and nodal tissues could be used as responsive explants, regardless of whether they are excised from seedlings or mature hardwood trees. We have extensive experience in using shoot tips (comprising the apical meristem and three to four leaf primordia with their axillary zones) and the two most apical expanding leaves to initiate SE in different *Quercus* and other hardwood species [28]. However, only shoot tips responded as suitable explants for initiating SE in holm oak. Shoot tips contain pluri- or toti-potent stem cells (the precursors of plant organs), which can favour SE induction [59,60]. The lack of response of holm oak leaf explants may be related to the tendency of the excised leaves to undergo rapid necrosis after culture for 48 h [33].

High concentrations of auxin with or without a cytokinin are generally used for initiating SE in non-zygotic explants [28]. The presence of auxins in the culture medium and the concentrations required to induce SE seem to be related to the level of cell differentiation of the initial explant. When these tissues contain predetermined embryogenic cells, somatic embryos are frequently initiated

on medium containing a weak auxin, such as IAA, or on medium devoid of PGRs. The results reported here are consistent with this approach, as SE was induced in medium supplemented with IAA and also medium without PGRs. This appears to corroborate the low level of differentiation of holm oak shoot apex, together with the excision stress of the explant tissues, which may be sufficient for redirecting the embryogenic pathway. According to [61], stress induces remodeling of the plant cell fate, with wound stress being a primary trigger for SE induction [62].

Maintenance of embryogenic competence in holm oak is considered problematic, as isolated somatic embryos at torpedo and cotyledonary-stages are not capable of generating new somatic embryos by secondary embryogenesis [19,26,32]. In the present study, several embryogenic lines from the two responsive genotypes were multiplied indefinitely and maintained in culture by using NSs as the explants to be subcultured for embryo proliferation. The use of NSs to generate somatic embryos was also investigated in embryogenic systems initiated from shoot apex explants derived from *Q. ilex* trees [33] and other related species, such as *Castanea dentata* [63], *C. sativa* [64], and *Q. rubra* [65]. Histological studies performed in NSs of white oak and red oak [65] revealed the presence of embryogenic cells capable of detaching from the surface layers and of proliferating as proembryonic aggregates, which may explain the high productivity of the NSs.

After a stratification period and subsequent transfer to germination medium, about 46% of somatic embryos converted into viable plants, indicating that SE is an additional method for micropropagating *P. cinnamomi*-tolerant *Q. ilex* plants. Further research is required specifically in the maturation and germination steps to improve the conversion rates obtained.

In conclusion, the results of the present study clearly show that the application of biotechnological approaches is a realistic possibility for the vegetative propagation of selected holm oak trees tolerant to *P. cinnamomi*. The two proposed procedures, i.e., axillary shoot proliferation and SE, enable the propagation of viable plants, although several problems must be solved. It is not clear which is the most appropriate procedure, although axillary shoot micropropagation can be considered technologically easier than SE. In addition, in order to initiate SE following the procedure described here, axillary shoot cultures are required as the original source of explants. However, once the embryogenic cultures are established, their multiplication capacity is generally higher than that of axillary shoots. In addition, two results derived from the two procedures used in the present study should be highlighted. First, plants obtained from the germination of somatic embryos have a pivotal root system (Figure 3D) similar to that of plants obtained from the germination of zygotic embryos (and acorns). By contrast, the rooted microshoots exhibit the typical arrangement of adventitious roots around the shoot (Figure 1E). Given the limited water availability in Mediterranean areas, a deep and pivotal root system may provide oak trees with a key advantage to survive drought conditions during summer [66]. Second, the medium-term conservation of holm oak germplasm has been demonstrated in axillary shoot cultures. However, cryopreservation under liquid nitrogen is ideal for long-term germplasm conservation. Our experience in cryopreservation of embryogenic cultures of different Fagaceae species [67–70], including *Q. ilex* [19], highlights the good potential offered by this technique for the long-term conservation of tolerant holm oak germplasm [67–70]. After the successful cryopreservation of embryogenic lines derived from *Q. ilex* mature trees, a positive response of holm oak somatic embryos developed from tolerant genotypes (being studied, at present) is also expected. Cryopreservation of embryogenic lines will be a valuable tool during the field testing of regenerated plants following a similar model to that of multi-varietal forestry defined for conifer species and the use of tested tree varieties in plantation forestry [54,71].

The micropropagation procedures defined in this study should also be tested by applying temporary immersion systems for large-scale propagation of these species, which would enable the production of *P. cinnamomi*-tolerant holm oak plants for use in future restoration programs in areas affected by the pathogen. In recent years, temporary immersion systems, such as RITA®, have been successfully applied in the proliferation of axillary shoots of chestnut [72], as well as in the proliferation of both pedunculate oak [73] and cork oak [74] embryogenic cultures.

**Supplementary Materials:** The following information is available online at http://www.mdpi.com/1999-4907/11/8/841/s1, **Supplementary information S1:** Supplementary **S1A.** Details of *Quercus ilex* stands used to select tolerant plants; **S1B.** Distribution of *Q. ilex* ssp. ballota (black circles) and *Q. ilex* ssp. ilex (orange circles) populations used in this study: (BR) Braganza, Tras-os-Montes e Alto Douro, Portugal; (BB) Barrios de Bureba, Castile and León, Spain; (BG) Bagnaia, Lazio, Italy. **Supplementary information S2:** Complementary information about the establishment step. A. Appearance of new shoots after forced sprouting in the growth cabinet. B. Flushed shoots removed from the plant shown in A before sterilization. C. Nodal explants used as initial explants for the establishment of in vitro cultures obtained from shoots shown in B. **Supplementary information S3:** Complementary information about the culture media used in the different steps of micropropagation by somatic embryogenesis and axillary shoot proliferation of holm oak.

**Author Contributions:** E.C. and M.T.M.: designed the study and participated in the research; N.R.-M.: participated in the research; F.J.V.: supervised the research; A.S. and R.T.: obtained tolerant holm oak plants. All authors have read and approved the final manuscript.

**Funding:** This research was partly funded by Xunta de Galicia (Spain) through the project Contrato Programa 2018–2019. Selection of UHu genotypes was funded by the Andalusian Regional Ministry of Science, Innovation and Busi-ness (project P07-RNM-03108), the European Social Fund and European Regional Development Fund.

**Acknowledgments:** We thank A.M. Vieitez and A. Ballester for comments and suggestions on the original manuscript. Technical assistance from Fátima Mosteiro is greatly appreciated. Eneko Pérez, Marilia Horta Jung, Anna Maria Vettraino, Alessia Tomassini and Andrea Vannini are acknowledged for their help in acorn collection.

**Conflicts of Interest:** The authors declare no conflicts of interest.

## Abbreviations

| | |
|---|---|
| AC | activated charcoal; |
| BA | benzyladenine |
| GD | Gresshoff and Doy medium |
| IAA | indole-acetic acid |
| IBA | indol-3-butyric acid |
| MS | Murashige and Skoog medium |
| NAA | $\alpha$-naphthalene acetic acid |
| NSs | nodular embryogenic structures |
| SE | somatic embryogenesis |
| SH | Schenk and Hildebrandt |
| STS | silver thiosulphate |
| UEx | University of Extremadura |
| UHu | University of Huelva |
| WPM | Woody Plant Medium |

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
