# Peer review of "Vegetative Propagation of Phytophthora cinnamomi-Tolerant Holm Oak Genotypes by Axillary Budding and Somatic Embryogenesis"

_forests, doi:10.3390/f11080841_

Round 1

Reviewer 1 Report

The presented study is very interesting and may have true value in practical use in the propagation of Phytophthora cinnamomi-tolerant holm oak genotypes. However, there are some things which can be improved.

Point 1. In my opinion, the Abstract section has too many details. It does not make sense to write the medium ingredients in Abstract which is intended to encourage the readers to read the whole paper.

Point 2. In the Material and Methods section the description of all mediums and PGR preparation should be detailed with sterilization procedure, source of the ingredients (commercial), how and when the PGRs have been sterilized, etc.  A good solution would be making a table with a composition of all used in the study mediums, pH, and other “specific” steps which can be crucial.

Point 3. I have objections about the number of repeats in the performed experiments. Should be at least three not two. Could you perform additional at least one? I understand that the material has been limited but at least authors should mark somehow in which case they have 2 repeats and in which more. The authors did not mention about repeats in the cold storage of shoots analysis.

Point 4. On page 4  line 47, there are some mistakes or additional words.

Point 5. The data presentation in tables is unintuitive. That way to present the statistical significance is unclear and should be reconsidered. In my opinion, marked significantly different results by * (star) or another symbol which clear description in the legend what to what is compered, would be much better.

 Point 6. On page 6 line 3 ‘did not vary greatly’ is not a scientific term. Was the difference statistically significant or not?

Point 7. Authors use in the manuscript term ‘expression medium’. Does it mean induction medium? ‘Expression’ should be rather used to describe gene expression. Moreover on page 7 line 5, page 8 line 7, page 9 line 47 authors should define what kind of medium it is ‘root? Induction/expression medium’?

Point 8. Page 7 line 42 does not have sense.

Point 9. Did the authors check the polarity of the somatic embryo? As they ‘somatic embryos were clearly bipolar’. The results from somatic embryogenesis induction analyses should be presented in a table, especially the results from analyses of maintenance of embryogenic competence. The statement that for example ‘developmental stage (…) greatly influenced embryo production …’ is not enough. The authors should show the results in the table with performed statistical analyses.

Point 10. In the discussion on page 10 line 18; are there available protocols for somatic embryogenesis induction for other subspecies and within it rotudifolia?

Point 11. The abbreviation list could be helpful for the readers.

Author Response

In the attachment, you can find a new version of manuscript and our answers.

Reviewer 2 Report

The manuscript is clear and understandable written, guiding the reader through the whole research, beginning with the problem definition, the way of experimentation, the interpretation of the results, until the conclusions.

This is the first study reporting axillary shoot proliferation and conservation by slow growth storage in holm oak plants that are tolerant to the root-rot pathogen P. cinnamomi, specially on Quercus ilex.

There is need for some minor corrections in manuscript:

  1. In vitro should be written in italics. Please, correct in the whole manuscript.
  2. Chapter: Results,

page 6, line 7: PR-8 should be corrected to PR8; the value 71.5 should be corrected to 71.4

line 34: PR-11 should be corrected to  PR11

line 43: (0-12%) should be corrected to (0-19.4%)

page 7, lines 1 and 43: PR-11 should be corrected to PR11

            line 48: PR-9 should be corrected to PR9

Chapter: Discussion,

          Page 22, lines 1 and 3: PR-11 should be corrected to PR11

  1. Table 2: The heads of the 3rd and 4th column of the table should be better formatted and clearly composed. Eg, word ‘’shoots’’ should be in upper part of the cell, and “ 0.5-1 cm (NO)'' below, in the lower part of the same cell of the table.
  2. Table 4: in the footnote of the Table 4, change AIB into IBA

Round 2

Reviewer 1 Report

I would recommend the paper in present form for publishing after careful re-reading by the authors themselves to avoid long sentences and unnecessary words.